# Interrelationship Between Cardiopulmonary Exercise Testing Indices and Markers of Subclinical Cardiovascular Dysfunction in Those with Type 2 Diabetes—An Observational Cross-Sectional Analysis

**DOI:** 10.3390/jfmk10040371

**Published:** 2025-09-26

**Authors:** Grace W. M. Walters, Gaurav S. Gulsin, Joseph Henson, Stavroula Argyridou, Kelly S. Parke, Thomas Yates, Melanie J. Davies, Gerry P. McCann, Emer M. Brady

**Affiliations:** 1Department of Cardiovascular Sciences, University of Leicester and the National Institute for Health Research (NIHR) Leicester Biomedical Research Centre, Leicester LE3 9QP, UK; grace.walters@ntu.ac.uk (G.W.M.W.); gg149@leicester.ac.uk (G.S.G.); kelly.parke@uhl-tr.nhs.uk (K.S.P.); gpm12@leicester.ac.uk (G.P.M.); 2Sport, Health, and Performance Enhancement (SHAPE) Research Centre, Department of Sport Science, School of Science and Technology, Nottingham Trent University, Nottingham NG11 8NS, UK; 3Diabetes Research Centre, University of Leicester and the NIHR Leicester Biomedical Research Centre, Leicester LE5 4PW, UK; jjh18@leicester.ac.uk (J.H.); argyridou.s@gmail.com (S.A.); ty20@leicester.ac.uk (T.Y.); melanie.davies39@nhs.net (M.J.D.)

**Keywords:** Cardiac magnetic resonance imaging, exercise tolerance, heart failure, type-2 diabetes

## Abstract

**Purpose**: While peak oxygen uptake ($\overset{.}{V}$O_2peak_) is the gold standard method for assessing exercise tolerance, there is a tendency for underestimation. Several other cardiopulmonary exercise testing (CPET) variables may provide additive prognostic value beyond $\overset{.}{V}$O_2peak_ alone. The aim of this study was to examine if alternative CPET indices of exercise tolerance are (a) impaired in people with T2D and (b) independently associated with measures of cardiovascular structure and function measured via echocardiography and cardiac MRI. **Methods**: Participants with type 2 diabetes (T2D) and healthy controls underwent cardiac magnetic resonance imaging, transthoracic echocardiography, and a CPET. Multiple linear regression was used to determine the relationship between indices of exercise tolerance and markers of cardiovascular structure and function. **Results**: A total of 84 people with T2D and 36 healthy volunteers were included in the analysis. All CPET outcomes were worse in those with T2D vs. the controls. Three CPET outcomes were associated with markers of cardiovascular structure and function: $\overset{.}{V}$O_2_ recovery with mean aortic distensibility (β = 0.218, *p* = 0.049); heart rate recovery with early filling velocity on transmitral Doppler/early relaxation velocity (β = −0.270, *p* = 0.024), left ventricular mass/volume ratio (β = −0.248, *p* = 0.030) and mean aortic distensibility (β = 0.222, *p* = 0.029); and $\overset{.}{V}$O_2_ at the ventilatory threshold with myocardial perfusion reserve (β = 0.273, *p* = 0.018). **Perspective**: These lesser-used CPET indices could be used to identify which people with T2D are at elevated risk of progression to symptomatic heart failure. However, larger longitudinal studies are required to confirm these findings and their potential clinical application.

## 1. Introduction

People with type 2 diabetes (T2D) have more than a two-fold increased risk of developing heart failure (HF) compared to those without T2D [1,2]. Prior to the development of symptomatic HF, up to half of people with T2D have asymptomatic left ventricle (LV) dysfunction [3,4], meeting the classification for stage B HF [5].

A reduced exercise tolerance is a hallmark of HF [6]. Numerous studies have evaluated the cardiopulmonary performance in response to physical exercise in T2D subjects and consistently observed a reduced peak oxygen uptake capacity ($\overset{.}{V}$O_2peak_) in comparison to non-T2D subjects [7]. A reduced exercise tolerance, expressed either by a reduced peak workload or $\overset{.}{V}$O_2peak_, is a powerful marker of an impaired global health status as well as a hallmark of HF [6]. People with T2D who are asymptomatic of any cardiovascular disease have been reported to have a 20–30% reduction in $\overset{.}{V}$O_2peak_ [8,9,10,11] and an early diabetes-related cardiopulmonary impairment has been postulated [12].

Cardiopulmonary exercise testing (CPET) is considered the gold standard non-invasive measure for assessing cardiorespiratory fitness and exercise tolerance [13]. CPET provides a simultaneous assessment of the cardiovascular, respiratory, muscular, and metabolic systems [14,15,16]. While $\overset{.}{V}$O_2peak_ bears strong prognostic value in several patient populations, including people with T2D [17], there is a tendency for $\overset{.}{V}$O_2peak_ to be underestimated because of reduced patient motivation and premature termination of the exercise test by the examiner, or by the patient, due to feeling uncomfortable and unfamiliar with the level of physical exertion. More recently, other CPET parameters have emerged that offer insight into multiorgan physiologic reserve capacity and provide better or additive prognostic value than $\overset{.}{V}$O_2peak_ [18,19,20]. This includes the following: the cardiorespiratory optimal point (COP) [21,22], HR reserve [23], the ventilatory threshold 24, oxygen uptake recovery ($\overset{.}{V}$O_2_ recovery) [24], and HR recovery [25].

Given the tendency for $\overset{.}{V}$O_2peak_ to be underestimated alongside the more recently examined CPET parameters that offer insight into multiorgan physiologic reserve capacity and provide better or additive prognostic value than $\overset{.}{V}$O_2peak_, there is a need to explore whether these alternative CPET indices of exercise tolerance are worse among people with T2D and obesity, and whether any are associated with markers of cardiovascular remodelling and dysfunction. The research gap this study aims to fill is whether there is any evidence that alternative CPET indices of exercise tolerance could hold any prognostic value for cardiovascular remodelling and dysfunction. By understanding whether these alternative CPET indices of exercise tolerance are worse in people with T2D and obesity, and whether there are any associations that exist between cardiovascular remodelling and dysfunction, further research can begin to explore whether these alternative CPET indices of exercise tolerance hold any possible prognostic value for people with T2D and obesity.

Therefore, the aim of this analysis was to examine if alternative parameters of exercise tolerance measured via CPET are (a) impaired in people with T2D without cardiovascular disease compared to age-, sex-, and ethnicity-matched healthy controls and (b) if they are independently associated with measures of cardiovascular structure and function (dependant variable), specifically early filling velocity on transmitral Doppler/early relaxation velocity [E/e′], LV mass/volume ratio, aortic distensibility [AoD], peak early diastolic strain rate [PEDSR], or myocardial perfusion reserve [MPR]. These markers of cardiovascular structure and function were selected because they have been reported to be abnormal in people with obesity and T2D without a history of and asymptomatic of any cardiovascular disease or dysfunction [26]. The hypothesis was that all alternative CPET indices of exercise tolerance measured would be worse in people with T2D and obesity compared to healthy matched controls, and that at least one of these alternative CPET indices of exercise tolerance would be associated with at least one measure of cardiovascular structure and function.

## 2. Methods

This is a secondary analysis of the “Diabetes Interventional Assessment of Slimming or Training to Lessen Inconspicuous Cardiovascular Dysfunction” (the DIASTOLIC trial) [26]. The methods of the DIASTOLIC trial have been previously described in detail [26]. In summary, the DIASTOLIC trial was a single-centre study, which comprised a baseline cross-sectional case–control analysis in adults with T2D compared to age-, sex-, and ethnicity-matched non-T2D controls.

### 2.1. Participants

Participants with T2D were eligible if they were working-age adults (≥18 and ≤65 years) with established T2D and a body mass index (BMI) > 30 kg/m^2^ (or >27 kg/m^2^ if South Asian), but had no signs, symptoms, or history of any cardiovascular disease or dysfunction (asymptomatic of any cardiovascular disease or disorder). Ethical approval was granted by the National Research Ethics Service. The study’s sponsor was the University of Leicester (Research Ethics Committee West Midlands–Coventry, reference: 15/WM/0222, 7 July 2015). The study was conducted in accordance with the International Conference on Harmonisation-Good Clinical Practice guidelines and the Declaration of Helsinki. All participants provided written informed consent in advance of entering the study.

### 2.2. Assessments

All participants underwent the following assessments: demographics, medical history, anthropometric measures, fasting blood samples, comprehensive contrast-enhanced, stress and rest perfusion cardiac magnetic resonance imaging (CMR), transthoracic echocardiography, and a symptom-limited cardiopulmonary exercise test (CPET).

### 2.3. Comprehensive Contrast-Enhanced, Stress and Rest Perfusion Cardiac Magnetic Resonance Imaging

CMR scanning was performed using a standardised protocol on a Siemens scanner (Erlangen, Germany) at 1.5T (Siemens Aera) with an 18-channel cardiac coil and retrospective electrocardiographic (ECG) gating. This standardised scanning protocol has been described in detail previously [27]. Patients were advised to abstain from caffeine-containing products at least 12 h prior to CMR scanning. CMR images were analysed offline while blinded to all patient details and treatment groups. Cardiac chamber volume, function, and strain were assessed by a single experienced observer (G.S.G.) using cmr42 version 5 (Circle Cardiovascular Imaging, Calgary, Alberta, Canada). Aortic distensibility was analysed by two experienced operators using Java Image Manipulation version 6 (Xinapse Software, Essex, UK).

### 2.4. Transthoracic Echocardiography

Comprehensive transthoracic echocardiography was performed according to the British Society of Echocardiography guidelines [28] by one of three accredited operators. Each participant was assessed for early diastolic transmittal flow velocity (E) and early diastolic mitral annular velocity (e′) to estimate LV filling pressures via Doppler echocardiography as E/e′ is a non-invasive surrogate for LV filling pressure [29].

### 2.5. Symptom-Limited Cardiopulmonary Exercise Test

Participants underwent a symptom-limited maximum incremental exercise test on a stationary bicycle (electromagnetically-braked cycle ergometer) with expired gas analysis to determine $\overset{.}{V}$O_2_peak. The exercise test began with a 3 min warm-up phase with 0 watts of resistance. Participants’ resistance then increased each minute by a pre-determined workload which had been calculated based on participants’ age, sex, height and weight using the following calculation [30]:
$$\text{Work Rate Increment }(\text{Watts}/\text{min})=(\overset{.}{V}O_{2\text{peak}}\;-\;\overset{.}{V}O_{2}\text{ unloaded})/100\;\overset{.}{V}O_{2\text{peak}}\;[\text{ml}/\text{min}]\;=\;\text{Females}:\;(\text{Height }[\text{cm}]\;-\text{ age})\;*\;14\;\text{Males}:\;(\text{Height }[\text{cm}]\;-\text{ age})\;*\;20\;\overset{.}{V}O_{2}\text{ unloaded }=\;150\;+\;(6\;*\text{ weight }[\text{kg}])$$

Before commencing the test, the air conditioning temperature was set to 19 degrees Celsius. Prior to test initiation, a brief description of the test was given to participants. It was emphasised to participants that they were in charge and could stop if they felt distressed, but that it was important for them to exercise to their maximum level. Participants were also taught to use a ‘thumb up’ signal to signify that everything was satisfactory and a ‘thumb down’ if they were experiencing any difficulty but did not wish to stop. Participants were also advised to point to the site of discomfort, such as the chest or legs.

Participants were then prepared for testing; the skin was prepped to attach the ECG electrodes, and a blood pressure cuff and oximeter were attached. It was explained to participants that they should aim for 60–80 revolutions per minute, and the process of the incremental increase in resistance was explained. It was re-emphasised to participants that they should exercise until they felt that they were unable to continue.

Once the participant was confident with the protocol, their position on the bike was checked to ensure comfort and correct technique, and the face mask was placed on the participant’s face. Once the participant was comfortable, the exercise test was initiated.

Once the participant indicated that they were at maximal exertion and that they were unable to continue, the test was terminated.

The raw data from each CPET test were exported to an excel sheet and were then later used for the manual calculation of each of the several CPET variables. A description of how each CPET variable was calculated is presented in Table 1.

### 2.6. Statistical Analysis

Statistical analyses were performed using a commercially available software package (IBM SPSS Statistics; version 26. Hampshire, UK). Normality of the outcome measures was assessed using histograms, the Shapiro–Wilk test, and Q-Q plots. To compare CPET indices of exercise tolerance between people with obesity and T2D and age-, sex-, and ethnicity-matched healthy controls, independent samples *t*-tests or Mann–Whitney tests were employed depending on data distribution. To examine the association between CPET indices of exercise tolerance and markers of cardiovascular structure and function, a multiple linear regression model was used to assess whether each CPET variable was associated with each marker of cardiovascular structure and function. This model was adjusted for age, sex, and smoking status. These parameters were adjusted for as they were reported to be major determinants of subclinical cardiovascular dysfunction in this cohort of working-age adults with T2D [26]. An exploratory analysis which included adjusted for BMI was also run to assess the impact of body mass (Appendix A). This exploratory analysis was performed on both people with T2D and healthy volunteers. Results for the associations between the CPET variables and cardiovascular structure/function were further adjusted for multiple testing (Bonferroni correction). A *p*-value of 0.05 was considered statistically significant.

## 3. Results

### 3.1. Participant Characteristics

A total of 84 people with T2D and obesity and 36 healthy volunteers had completed a CPET and were therefore included in the case vs. control baseline analysis of CPET indices of exercise tolerance. The mean T2D duration for those with T2D was 62.4 ± 38.2 months. The healthy volunteers (controls) and T2D with obesity group (cases) were similar for age, sex, and ethnicity distribution; however, as expected, the control group had lower overall body weight, BMI, blood pressure, HR, and HbA1c (Table 2).

### 3.2. Cardiovascular Structure and Function

#### Cardiac Magnetic Resonance Imaging

The values for the baseline CMR and echocardiographic imaging in those with T2D vs. controls are shown in Table 1 and have been described in detail previously [26]. Briefly, the CMR revealed that compared to controls, those with T2D had a significantly lower LV peak early diastolic strain rate (1.10 ± 0.16 vs. 1.01 ± 0.19, respectively, *p* = 0.02), smaller indexed LV volumes, higher LV ejection fraction, higher LV mass, increased concentric LV remodelling (LV mass: volume, 0.71 ± 0.10 vs. 0.82 ± 0.12 g/mL, respectively, *p* < 0.001) and lower mean aortic distensibility (6.56 ± 2.02 vs. 4.16 ± 2.05 mmHg^−1^ × 10^−3^, respectively, *p* < 0.001). Those with T2D also had a lower myocardial perfusion reserve.

### 3.3. Echocardiography

Compared to healthy matched controls, those with T2D had a significantly lower mean E/A (1.21 ± 0.25 vs. 0.95 ± 0.21, respectively, *p* < 0.001) and a significantly higher E/e′ (6.2 [5.0–7.8] vs. 8.1 [6.2–9.6], respectively, *p* < 0.001) (Table 2).

### 3.4. Indices of Exercise Tolerance

All assessed CPET outcomes were significantly impaired in those with T2D compared to controls (Table 3). HR reserve was >50% higher in controls (*p* < 0.001), meaning that healthy volunteers had the ability to increase their HR by more than 50% compared to people with obesity and T2D. $\overset{.}{V}$O_2_ at the ventilatory threshold ($\overset{.}{V}$O_2_VT), a marker of aerobic capacity, was 38% lower in people with T2D (*p* < 0.001) demonstrating that the ventilatory threshold is achieved earlier in people with obesity and T2D compared to their healthy counterparts. SlopeV_E_/V_CO2_ was 10% (*p* = 0.008) and COP was 6% lower in controls (*p* = 0.009) compared to T2D. $\overset{.}{V}$O_2_ recovery was 14% faster in healthy volunteers (*p* < 0.001), demonstrating a delay in $\overset{.}{V}$O_2_ recovery in people with obesity and T2D. HR recovery was 4% faster in healthy volunteers (*p* < 0.001).

### 3.5. Associations with Markers of Cardiovascular Remodelling

The relationship between the three alternative parameters of exercise tolerance and five cardiovascular structure/function on CMR in people with obesity and T2D, with adjustment for multiple testing, are provided in Table 4 and described here. $\overset{.}{V}$O_2_ recovery was significantly positively associated with mean aortic distensibility (β = 0.218, *p* = 0.049), demonstrating that a faster $\overset{.}{V}$O_2_ recovery was associated with less aortic stiffness. HR recovery had a significant inverse association with E/e′ (β = −0.270, *p* = 0.024), indicating that a slower HR recovery was associated with elevated E/e′ (higher LV filling pressures). HR recovery had a significant negative association with LV mass/volume ratio (β = −0.248, *p* = 0.030), suggesting a slower HR recovery was associated with elevated LV mass/volume ratio (more concentric remodelling). HR recovery had a significant positive association with mean aortic distensibility (β = 0.222, *p* = 0.029) indicating that a faster HR recovery was associated with increased aortic distensibility (less aortic stiffening). Finally, $\overset{.}{V}$O_2_VT was significantly positively associated with MPR (β = 0.273, *p* = 0.018), thus suggesting that a higher $\overset{.}{V}$O_2_VT (the ventilatory threshold occurs later in the exercise course) was associated with a higher MPR, representing less microvascular dysfunction.

Healthy volunteers showed no association between alternative parameters of exercise tolerance and the five cardiovascular structure/function on CMR (Appendix A).

## 4. Discussion

To our knowledge, this is the first study to compare alternative parameters of exercise tolerance markers derived from CPET in people with obesity and T2D with evidence of subclinical cardiovascular structural and functional abnormalities that align with stage B HF vs. healthy controls. In addition, this is also the first study to assess the association between indices of exercise tolerance, beyond $\overset{.}{V}$O_2peak_ and cardiovascular structure/function in this patient population.

### 4.1. Key Findings

This secondary analysis of the DIASTOLIC trial [26] revealed two key findings: Firstly, all the CPET indices of exercise tolerance assessed, which included ventilatory efficiency (SlopeV_E_/V_CO2_), $\overset{.}{V}$O_2_ recovery, HR recovery, HR reserve, $\overset{.}{V}$O_2_VT, and COP, are impaired in people with obesity and T2D compared to age-, sex-, and ethnicity-matched healthy controls. Secondly, $\overset{.}{V}$O_2_ recovery, HR recovery, and $\overset{.}{V}$O_2_VT all revealed significant associations with markers of cardiovascular structure/function independent of age, sex, and smoking status. A visual summary of the key findings is presented in Figure 1.

While it is well recognised that people with T2D have a reduced exercise capacity [31], this has almost exclusively been demonstrated using $\overset{.}{V}$O_2peak_ as the marker of exercise tolerance, but there is an array of CPET outcomes which have shown prognostic power in HF, yet exploration into how these outcomes compare in people with T2D, and whether these indices are associated with abnormal cardiovascular structure and function, for the most part, has been neglected. These alternative indices offer an opportunity to potentially detect early determinants of progression to HF in people with T2D that could be modifiable [32]. 

### 4.2. $\overset{.}{V}$O_2_ at the Ventilatory Threshold

It has been suggested that an early ventilatory threshold is associated with a higher risk of death in people with HF [33], with a ventilatory threshold of <11 $\overset{.}{V}$O_2 mL_/kg/min suggested as the threshold for those with a high risk of HF-related mortality. $\overset{.}{V}$O_2_VT has also demonstrated greater prognostic strength for predicting 6-month mortality than $\overset{.}{V}$O_2peak_ [33]. The mean $\overset{.}{V}$O_2_VT in the present investigation in those with obesity and T2D was 9.1 mL/kg/min and, on an individual level, 62 (73%) of people with obesity and T2D fell below the threshold of a $\overset{.}{V}$O_2_VT < 11, while only 7 (19%) of healthy volunteers met this criterion. Thus, this suggests that people with obesity and T2D, despite being asymptomatic of any cardiovascular dysfunction, may be at increased risk of HF-related mortality.

Linear regression modelling revealed an association between $\overset{.}{V}$O_2_VT and MPR. $\overset{.}{V}$O_2_VT showed a positive relationship with MPR (a measure of microvascular function) which is defined as the maximal increase in myocardial blood flow above rest expressed as a ratio to resting blood flow. A reduced MPR has previously been shown to associate with LV diastolic dysfunction [34] and has frequently been reported as being reduced in people with T2D [34,35,36]. Given that diminished MPR has also been frequently observed in patients with chronic HF [37,38,39,40,41,42], one may postulate that those who are at higher risk of developing symptomatic HF may be identified using $\overset{.}{V}$O_2_VT from CPET, but longitudinal studies are required to confirm this hypothesis.

### 4.3. $\overset{.}{V}$O_2_ Recovery

Oxygen uptake and HR recovery following exercise both predict adverse outcomes in HF [43,44]. $\overset{.}{V}$O_2_ recovery in the 60 s following exercise in the present study was reduced in people with obesity and T2D compared to age-, sex-, and ethnicity-matched healthy controls, demonstrating a delayed $\overset{.}{V}$O_2_ recovery in obese T2D participants. Abnormally prolonged $\overset{.}{V}$O_2_ recovery following exercise has been observed in patients with HF [45,46] and was linked to higher mortality rates in 200 chronic HF patients [44]. The current study adds to the depth of evidence on reduced $\overset{.}{V}$O_2_ kinetics in T2D, but more specifically in patients with stage A and B HF.

It was also observed that $\overset{.}{V}$O_2_ recovery had a positive association with mean AoD. Associations between proximal aortic stiffening, as indicated by decreased aortic distensibility, and HF development have been observed in patients with asymptomatic diastolic dysfunction [47]. LV mass and aortic stiffening are well established predictors of adverse cardiovascular outcomes [48]. We have previously demonstrated that AoD is independently associated with concentric LV remodelling in younger asymptomatic adults with T2D [49]. This supports the postulation that one mechanism through which aortic stiffening drives adverse cardiovascular outcomes in the context of T2D is via adverse LV remodelling, which increases the risk of subsequent HF [49]. Notably, our T2D participants had a 20% increase in concentric LV remodelling compared to matched healthy controls [49]. Targeting aortic stiffness (and by proxy LV hypertrophy) could potentially prevent progression to overt, symptomatic HF, which is of particular importance in younger T2D subjects, who have the highest lifetime risk of adverse cardiovascular outcomes, and in whom aortic stiffening and cardiac remodelling is likely to be reversible. However, in this present analysis, $\overset{.}{V}$O_2_ recovery did not associate with LV mass/volume ratio (a marker of concentric remodelling).

### 4.4. Heart Rate Recovery

In the present study, HR recovery was also reduced in people with obesity and T2D. HR recovery was also associated with a reduced AoD, along with a higher LV mass/volume ratio and elevated E/e′. Reduced HR recovery is a very powerful predictor of increased mortality in obesity [50] in apparently healthy people [51,52,53,54,55] and in people with HF [25,43,56,57,58,59]. Impaired HR recovery is highly prevalent in asymptomatic T2D subjects [60] and has been linked to adverse cardiovascular outcomes and increased all-cause mortality [61,62]. Reduced HR recovery after exercise may be a manifestation of cardiac autonomic neuropathy, which is a common complication of T2D. HR recovery was associated with an increased LV mass: volume (a marker of concentric remodelling) ratio and elevated E/e′ (an index to evaluate LV filling pressure/assess LV diastolic function) [63,64]. Concentric remodelling has been associated with impaired diastolic function [65]. HR recovery, therefore, appears to be an important CPET outcome in people with T2D that could be used to indicate which T2D patients are likely to develop significant cardiovascular structural/functional abnormalities.

### 4.5. Strengths and Limitations

This is the first study to investigate the associations between alternative parameters of exercise tolerance derived from CPET in people with T2D and obesity, and cardiovascular abnormalities aligned with stage B HF. This, however, is a very early suggestion and requires a significantly larger evidence base before we can make more substantial claims about the use of CPET for risk stratification within T2D patient cohorts. This investigation was in a relatively small cohort and had more than twice as many people with T2D compared to healthy volunteers. This investigation also looked at CPET indices during exercise and explored how they associate with cardiovascular structure and function at rest (except myocardial blood flow). Furthermore, the results of this study are purely observational, and therefore, no causality can be inferred or assumed from the results.

## 5. Perspective

Overall, it is also evident from the present investigation that there is an array of indices of exercise tolerance that can be obtained from CPET which demonstrate impaired exercise tolerance in people with obesity and T2D when compared to age-, sex-, and ethnicity-matched healthy controls, which offer additive or improved prognostic power compared to $\overset{.}{V}$O_2peak_. These findings may inform risk stratification approaches to the identification of those at elevated risk of progressing to stage C or D HF. However, larger longitudinal studies are required to confirm these findings and their potential clinical application. Future research on case–control comparisons should include larger sample sizes with even more matching (e.g., for weight and blood pressure).

## Figures and Tables

**Figure 1 jfmk-10-00371-f001:**
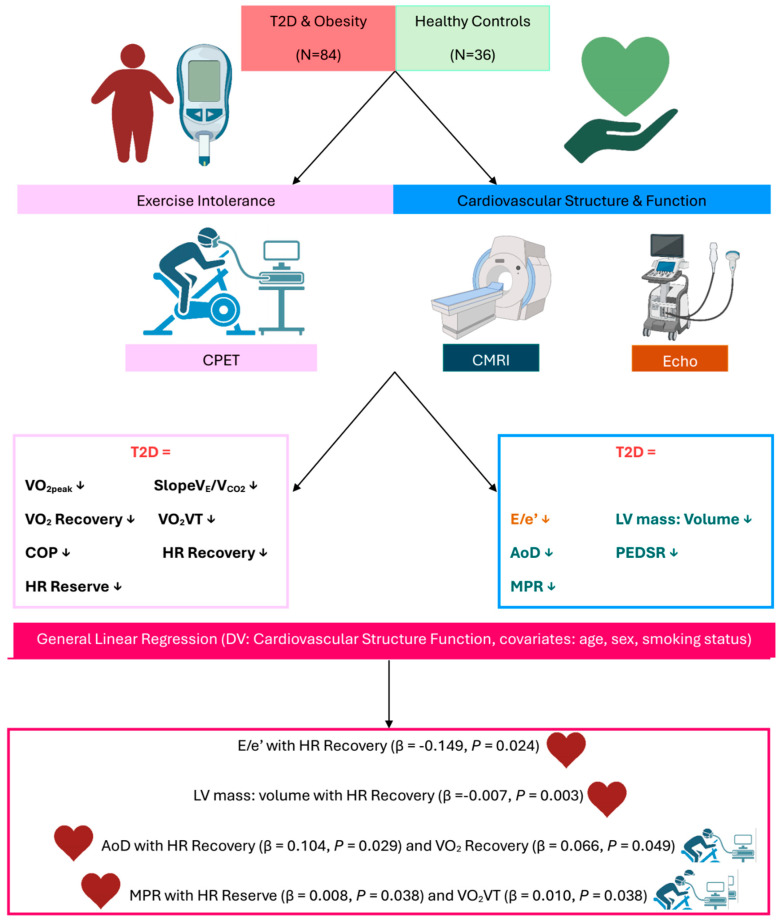
Visual summary of key findings. A visual summary of the key findings. Abbreviations: T2D: Type-2 diabetes; CPET: Cardiopulmonary exercise test; CMRI: Cardiac magnetic resonance imaging; Echo: Echocardiography; DV: dependant variable; V_E_: minute ventilation; V_CO2_: Carbon Dioxide Output; V_E_/V_CO2_: ventilatory efficiency; $\overset{.}{V}$O2: oxygen uptake; HR: heart rate; VT: ventilatory threshold; COP: cardiorespiratory optimal point; E/e′: Mitral inflow velocity/ mitral annular early diastolic velocity; LV: left ventricular.

**Table 1 jfmk-10-00371-t001:** Calculations for CPET indices of exercise tolerance.

CPET Indices of Exercise Tolerance	Calculation
SlopeV_E_/V_CO2_ slope	The linear regression slope of V_E_ and V_CO2_ from the start of the exercise to the VT. Presented as the gradient of the slope.
$\overset{.}{V}$O_2_ Recovery	How much $\overset{.}{V}$O_2_ falls from peak to 60 s post-exercise cessation. Presented as the percentage decrease from peak to 60 s post-exercise cessation.
Heart rate recovery	How much heart rate falls from peak to 60 s post-exercise cessation. Presented as the percentage decrease in heart rate from peak to 60 s post-exercise cessation.
Heart rate reserve	How much the heart rate increases from resting to peak exercise. Presented as the percentage increase from resting to peak exercise
Oxygen uptake at the ventilatory threshold	The $\overset{.}{V}$O_2_ (mL/kg/min) value at which the ventilatory threshold (VT) occurs. The VT was calculated using the V-Slope method.
Cardiorespiratory optimal point	The lowest value of V_E_/V_CO2_ in a given minute during an incremental exercise test. V_E_/V_CO2_ was averaged over every minute during the CPET and the lowest is the COP.

VE: minute ventilation; V_CO2_: carbon dioxide output; V_E_/V_CO2_: ventilatory efficiency; $\overset{.}{V}$O_2_: oxygen uptake; HR: heart rate; VT: ventilatory threshold; COP: cardiorespiratory optimal point.

**Table 2 jfmk-10-00371-t002:** Participant characteristics.

	T2D	Healthy Controls	*p*-Value
	**N = 84**	**N = 36**	
Demographics and anthropometry			
Age (years)	50.5 ± 6.3	48.6 ± 6.2	0.124
Sex (N [%] female)	34 [40%]	17 [47%]	0.466
Weight (kg)	102.2 ± 14.9	70.4 ± 10.8	<0.001
BMI (kg/m^2^)	36.5 ± 5.2	24.5 ± 2.4	<0.001
SBP (mmHg)	140.0 ± 16.5	120.94 ± 13.24	<0.001
DBP (mmHg)	87.7 ± 9.1	76.44 ± 7.15	<0.001
Resting HR (BPM)	73.9 ± 9.4	61.7 ± 9.8	<0.001
Smoking Status (N [Never/Ex/Current])	44/22/14	27/08/2001	0.027
Ethnicity (N [% BAME])	34 [40%]	12 [33%]	0.494
T2D Duration (months)	64.3 ± 38.7	N/A	N/A
Cardiovascular Structure/Function			
E/e′	8.3 ± 2.4	6.4 ± 1.5	<0.001
LV mass: volume (g/mL)	0.8 ± 0.1	0.7 ± 0.1	<0.001
Aortic Distensibility (mmHg^−1^ × 10^3^)	4.1 ± 2.1	6.6 ± 2.0	<0.001
Peak Early Diastolic Strain Rate (S^−1^)	1.0 ± 0.2	1.1 ± 0.2	0.008
Myocardial Perfusion Reserve	3.0 ± 0.9	3.9 ± 0.01	<0.001
T2D Control			
HbA1c (mmol/mol)	56 ± 11	35 ± 3	<0.001
HbA1c (%)	7.3 ± 10	5.4 ± 0.2	<0.001

Demographics, anthropometry, and cardiovascular structure and function variables in T2D vs. healthy volunteers. Data presented as mean ± standard deviation. The significance value is the difference between people with T2D and age-, sex-, and ethnicity-matched controls. Abbreviations: BPM: beats per minute, DBP: diastolic blood pressure, HbA1c: glycated haemoglobin, mmHg: millimetre of mercury, SBP: systolic blood pressure, T2D: type 2 diabetes, SD: standard deviation, CI: confidence interval.

**Table 3 jfmk-10-00371-t003:** CPET variables in T2D vs. healthy volunteers.

	T2D	Healthy Controls	
CPET Variable	N = 84	N = 36	*p*-value
$\overset{.}{V}$O_2_peak (L/min)	1.7 ± 0.5	1.9 ± 0.7	0.012
$\overset{.}{V}$O_2_peak (mL/kg/min)	16.6 ± 4.0	27.5 ± 8.6	<0.001
SlopeV_E_/V_CO2_	28.8 ± 5.4	26.1 ± 4.3	0.008
$\overset{.}{V}$O_2_ Recovery (%)	17.7 ± 6.7	20.4 ± 7.2	<0.001
$\overset{.}{V}$O_2_VT (mL/kg/min)	9.1 ± 2.6	13.3 ± 4.2	<0.001
Percentage of peak $\overset{.}{V}$O_2_ at VT (%)	54.9 ± 10.7	49.7 ± 9.5	0.012
COP	28.4 ± 3.1	26.7 ± 3.7	0.009
HR Reserve (%)	103.6 ± 32.6	154.9 ± 38.6	<0.001
HR Recovery (%)	12.3 ± 4.4	16.3 ± 5.5	<0.001
$\overset{.}{V}$O_2_ at VT < 11 (N [%])	62 [73]	12 [33]	<0.001
V_E_/V_CO2_ (<29.9) (N [%])	35 [41]	7 [19]	0.022

Cardiopulmonary exercise testing variables in T2D vs. healthy volunteers. Data presented as mean ± standard deviation. The significance value is the difference between people with T2D and age-, sex-, and ethnicity-matched controls. V_E_: minute ventilation; V_CO2_: carbon dioxide output; V_E_/V_CO2_: ventilatory efficiency; $\overset{.}{V}$O_2_: oxygen uptake; HR: heart rate; VT: ventilatory threshold; COP: cardiorespiratory optimal point; E/e′: mitral inflow velocity/mitral annular early diastolic velocity; LV: left ventricular.

**Table 4 jfmk-10-00371-t004:** Associations between novel CPET indices and cardiovascular structure/function.

	SlopeV_E_/V_CO2_	HR Recovery (%)	HR Reserve (%)	$\overset{.}{V}$O_2_VT (mL/kg/min)	$\overset{.}{V}$O_2_ Recovery (%)	COP
N = 84	E/e′					
β	0.023	−0.149	0.004	−0.027	−0.033	0.016
95% CI	−0.087, 0.133	−0.277, −0.02	−0.014, 0.022	−0.244, 0.189	−0.117, 0.05	−0.179, 0.211
*p*-Value	0.681	0.024	0.641	0.801	0.43	0.869
Effect size	0.003	0.072	0.003	0.001	0.009	<0.001
	LV mass: volume (g/mL)					
β	0.001	−0.007	−0.06	−0.031	−0.001	−0.002
95% CI	−0.005, 0.007	−0.014, −0.001	−0.001, 0.001	−0.011, 0.011	−0.006, 0.004	−0.012, 0.008
*p*-Value	0.738	0.030	0.818	0.988	0.695	0.671
Effect size	0.001	0.058	0.001	<0.001	0.002	0.002
	Mean Aortic Distensibility (mmHg^−1^ × 10^3^)					
β	0.041	0.104	0.008	−0.086	0.066	0.127
95% CI	−0.039, 0.121	0.011, 0.197	−0.005, 0.021	−0.244, 0.072	0.00, 0.132	−0.011, 0.266
*p*-Value	0.315	0.029	0.24	0.281	0.049	0.071
Effect size	0.013	0.059	0.017	0.015	0.05	0.041
	Peak Early Diastolic Strain Rate (S^−1^)					
β	0.002	0.009	−0.096	−0.009	−0.002	0.012
95% CI	−0.006, 0.009	−0.009, 0.009	−0.001, 0.002	−0.024, 0.006	−0.009, 0.004	−0.001, 0.025
*p*-Value	0.667	0.933	0.378	0.236	0.481	0.065
Effect size	0.002	<0.001	0.01	0.018	0.006	0.042
	Myocardial Perfusion Reserve					
β	0.001	0.036	0.008	0.1	−0.016	−0.035
95% CI	−0.043, 0.046	−0.017, 0.089	0.00, 0.016	0.017, 0.183	−0.054, 0.021	−0.114, 0.045
*p*-Value	0.949	0.182	0.038	0.018	0.388	0.386
Effect size	<0.001	0.026	0.063	0.08	0.011	0.011

Associations between novel CPET variables and cardiovascular structure and function. Data are derived from generalised linear modelling where each cardiovascular variable was selected as a dependent variable. Data are as β, 95% confidence interval (CI) of β, *p*-value. Effect size is partial eta squared. Abbreviations: V_E_: minute ventilation; V_CO2_: carbon dioxide output; V_E_/V_CO2_: ventilatory efficiency; $\overset{.}{V}$O_2_: oxygen uptake; HR: heart rate; VT: ventilatory threshold; COP: cardiorespiratory optimal point; E/e′: mitral inflow velocity/mitral annular early diastolic velocity; LV: left ventricular; AOD; aortic distensibility; PEDSR: peak early diastolic strain rate; MPR: myocardial perfusion reserve.

## Data Availability

The original contributions presented in this study are included in the article/Appendix A. Further inquiries can be directed to the corresponding author.

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
