# Peer review of "Interrelationship Between Cardiopulmonary Exercise Testing Indices and Markers of Subclinical Cardiovascular Dysfunction in Those with Type 2 Diabetes—An Observational Cross-Sectional Analysis"

_jfmk, 2025, doi:10.3390/jfmk10040371_

Round 1

Reviewer 1 Report

Comments and Suggestions for Authors

Peer review report

General Comments

The manuscript titled "Interrelationship Between Cardiopulmonary Exercise Testing Indices and Markers of Subclinical Cardiovascular Dysfunction in Those with Type 2 Diabetes" addresses a highly relevant topic in cardiometabolic health. This study is well structured and methodologically sound, which provides valuable information on early cardiovascular risk stratification in asymptomatic individuals with type 2 diabetes (T2D). Its novelty lies in the use of lesser known and/or used CPET variables, such as recovery of VOâ‚‚, recovery of HR and VOâ‚‚ at ventilatory threshold, and their association with subclinical cardiovascular abnormalities. The writing is for the most part clear, and the statistical approach is appropriate for the objectives that have been put forward. The study contributes significantly to the body of scientific literature focused on the early detection of heart failure (HF) in patients with T2D. The following are some recommendations for improvements:

Major adjustments

  1. Although multiple testing adjustment is mentioned, the manuscript does not clarify which correction method was applied, for example (e.g., Bonferroni). This should be explicitly stated.
  2. The discussion should more clearly emphasize the cross-sectional nature of the study when making inferences about possible clinical applications.

Minor adjustments

  1. Punctuation and grammar could be improved in some long or complex sentences to increase readability.
  2. Terminological inconsistencies need to be harmonized, for example:
    1. The alternation between “HR” and “heart rate” in all manuscript. The first time “HR” should be explained is on line 63.
    2. It is duplicated the CPET abbreviation in lines 50 and 93.
  3. In the line 62 please change “the aim of this analysis…” by “The aim of this study…”
  4. In line 114 please delete stationary bicycle and parentheses of “(electromagnetically braked cycle ergometer)”
  5. Line 115, At what temperature and humidity was the environment regulated?
  6. In line 147, values for “Those with T2D also had lower myocardial perfusion reserve”, are missing.
  7. In statistics, what level of significance was used, p<0.05????In Table 2 improve “SDev”.
  8. Figures or visual summaries of key findings (e.g., associations between CPET variables and cardiac indices) could help synthesize results more effectively.
  9. Tables should be improved. The legends are not equally formatted and in some cases state non-existent abbreviations such as SDIn line 254, [59 & 60]?
  10. In the strengths and limitations section, explicitly state that the results are observational and causality cannot be inferred.
  11. The list of references is extensive and well cited, but the format of the references should be double-checked to ensure consistency. Among others the “reference 5” cannot be accessed from the link. The webpage could not be found.

Final Recommendations

Perhaps it would be good to modify the title to a more concise and direct one, such as “Cardiopulmonary Exercise Testing Indices reflect Subclinical Cardiovascular Dysfunction in Type 2 Diabetes”.

Author Response

Overall comment

Thank you for the time taken to read this work and the incredibly helpful comments and suggested amendments. All comments and suggestions have been addressed with appropriate amendments made. The amended sections in the manuscript, tables, and figures have been highlighted in yellow for ease of identification.  We now feel that with the help of these comments and suggested amendments the quality of this manuscript has now been substantially improved. We would like to thank the reviewer again for the time taken to read and review this manuscript.

Please see below for responses to specific comments.

Responses to specific comments

Reviewer 1

  1. Comment: Although multiple testing adjustment is mentioned, the manuscript does not clarify which correction method was applied, for example (e.g., Bonferroni). This should be explicitly stated.

Answer: Thank you for identifying this missing information, this information has now been added.

  1. Comment: The discussion should more clearly emphasize the cross-sectional nature of the study when making inferences about possible clinical applications.

Answer: Thank you for this suggestion. This has now been addressed in the limitations section of the discussion.

  1. Comment: Punctuation and grammar could be improved in some long or complex sentences to increase readability.

Answer: Thank you for highlighting this. Punctuation and grammar have been amended throughout to improve the flow and readability of the manuscript.

  1. Comment: Terminological inconsistencies need to be harmonized, for example:

The alternation between “HR” and “heart rate” in all manuscript. The first time “HR” should be explained is on line 63. It is duplicated the CPET abbreviation in lines 50 and 93.

Answer: Thank you for highlighting this. This inconsistency has now been addressed throughout the manuscript to ensure consistency throughout.

  1. Comment: In the line 62 please change “the aim of this analysis…” by “The aim of this study…”

Answer: Thank you for your suggestion, this has been amended in line with your suggestion.

  1. Comment: In line 114 please delete stationary bicycle and parentheses of “(electromagnetically braked cycle ergometer)”

Answer: Thank you for this suggestion. This has been changed to reflect your suggestion.

  1. Comment: Line 115, At what temperature and humidity was the environment regulated?

Answer: Thank you for your comment highlighting this missing information. Additional information about how the CPET was conducted, including information regarding temperature and humidity has now been added to the results section.

  1. Comment 9: In statistics, what level of significance was used, p<0.05????In Table 2 improve “SDev”.

Answer: Thank you for highlighting this missing information. The level of significance has been added to the methods section and the wording in the tables regarding standard deviation has been amended.

  1. Comment: Figures or visual summaries of key findings (e.g., associations between CPET variables and cardiac indices) could help synthesize results more effectively.

Answer: Thank you for this suggestion. A visual summary of the key findings has been added.

  1. Comment: Tables should be improved. The legends are not equally formatted and in some cases state non-existent abbreviations such as SDIn line 254, [59 & 60]?

Answer: Thank you for your comment. All tables have been amended to improve clarity and consistency.

  1. Comment: In the strengths and limitations section, explicitly state that the results are observational and causality cannot be inferred.

Answer: Thank you for your comment. This has now been addressed and explicitly stated in the strengths and limitations section.

  1. Comment: The list of references is extensive and well cited, but the format of the references should be double-checked to ensure consistency. Among others the “reference 5” cannot be accessed from the link. The webpage could not be found.

Answer: Thank you for identifying this. The formatting of the references list has now been amended to ensure consistency throughout and to ensure all weblinks are accessible.

Reviewer 2 Report

Comments and Suggestions for Authors

Dear authors,

The present work is a well-written, straight-to-the-point manuscript focused on exploring the clinical relevance of “novel” CPET variables in individuals with type 2 diabetes by comparing them with healthy age-and-sex-matched individuals. They also explored the association between CPET variables and cardiovascular structure and function indices. The work is concise, features excellent data, and significantly contributes to the scientific knowledge by reporting the relationship of the CPET variables with TD2 individuals and potential insights on the incidence of heart failure in these patients. However, I believe several critical issues need to be addressed before acceptance.

MAJOR ISSUES/RECOMMENDATIONS

#

Topic

Comment

1

Results

A key aspect of a case-control study is to ensure that differences between groups are solely attributable to the condition (case). Because of this, the cases in this study have TD2 and obesity. Since healthy controls do not have obesity, and Obesity could influence all key variables of your study, I highly recommend including BMI adjustments in all of your analyses. This adjustment is critical to distinguish differences caused “solely” by TD2 from those caused by obesity and TD2. This adjustment can easily be done using an ANCOVA or a Generalized Linear Model analysis. Both can be done using SPSS.

Thus, authors would present a crude, unadjusted p-value and the adjusted p-value in their tables.

2

Results

Similar to the first comment, for the CPET analysis, I suggest presenting the absolute VO2 values alongside the relative ones, i.e., mL/min and mL/kg/min. Considering the striking differences in body weight, if one were to look at the VO2peak and VO2 at VT variables, it is remarkable that (using the average reported weight values) the difference between groups in terms of VO2peak would still be striking (1696 vs 1936 mL/min), but not in VO2 at VT (924 mL/min vs 938 mL/min).

3

Results

I think the findings of your multiple regression were extremely interesting, and I am eager to refer to them when published. However, since one of the aims was to understand this relationship within TD2 individuals, I recommend performing the same analyses for healthy individuals as well. Reporting this data would help determine whether the observed associations between CPET and cardiac structure/function are related to TD2/obesity pathophysiology or a common finding in your sample. Remember to check and report multivariate assumptions checks (e.g., independence and normality of errors).

4

Methods/
Results

As far as I am aware, the cardiorespiratory optimal point, especially considering the reference authors used in the introduction, is the lowest VE/VO2 during a CPET. Surprisingly, authors used the lowest VE/VCO2 instead. The latter is a better reflection of our ability to remove CO2 and respiratory efficiency, and the lowest VE/VO2 is a measure of circulation-respiratory “efficiency” as it represents the highest oxygen uptake for a 1L of ventilation.
Please address this issue.

5

General 

Please address medication use. Is this information available? Did the authors consider including it in the multiple regression? Such information is critical for HR recovery data. Please address this issue.

6

Methods

Please report critical information related to CPET “quality,” e.g., test termination criteria, peak RPE, peak RER, gas analyzer model/calibration/type (e.g., breath-by-breath vs mixed chamber), state pedaling frequency, room temperature (if controlled), etc.

7

Results

Sample sizes should be included in all tables and figures.

8

Statistics

The test used to compare both groups - I suppose it was a T-test - is not mentioned in section 2.6. This information is important to interpret findings reported in Tables 1 and 2. Also, please clearly state how the authors adjusted for multiple testing.

9

General

Often, the VE/VCO2 slope is presented solely as VE/VCO2, which, I believe, could be somewhat misleading. Please clarify this throughout the manuscript, clearly stating slope when appropriate, or, using another acronym like SlopeVE/VCO2.

10

Discussion

Revisit your limitations. Clearly address aspects such as the cross-sectional nature of the study, the inability to establish causation, the potential medication effect, the addictive nature of obesity, and T2D on the cardiovascular system.

11

Supplement

Since Table 1 contains critical information and there is no limit to the number of tables and figures, I strongly recommend including Table 1 in the main manuscript rather than as supplementary material.

Minor issues

#

Topic

Comment

1

General

Authors sometimes refer to the CPET variables analyzed as novel. I suggest different terms, as most of these variables have been investigated for more than ten years.

2

Results

Suggestion: Include effect sizes in your report. Since you have a large sample size, p-values can sometimes be significant even when differences are small.

3

Nomenclature

I recommend saying transmitral Doppler instead of transmittal.

4

Results

Please clarify if the βs used are standardized or not, and if they represent a change for every SD or unit.

5

Tables

Adjust decimal places for a relevant clinical/functional interpretation. Resting HR could be 88 or 87.7 instead of 87.68. Same thing for age, BP, etc.

6

General

Suggestion: Include a figure explaining your mechanistic rationale

Author Response

Overall comment

Thank you to the reviewer for the time taken to read this work and the incredibly helpful comments and suggested amendments. All comments and suggestions have been addressed with appropriate amendments made. The amended sections in the manuscript, tables, and figures have been highlighted in yellow for ease of identification.  We now feel that with the help of these comments and suggested amendments the quality of this manuscript has now been substantially improved. We would like to thank the reviewer again for the time taken to read and review this manuscript.

Please see below for responses to specific comments.

  1. Comment: I highly recommend including BMI adjustments in all of your analyses. This adjustment can easily be done using an ANCOVA or a Generalized Linear Model analysis. Both can be done using SPSS. Thus, authors would present a crude, unadjusted p-value and the adjusted p-value in their tables.

Answer: Thank you for your suggestion. This analysis has been run and the results have been included in a table in the supplementary materials.

  1. Comment: I suggest presenting the absolute VO2values alongside the relative ones.

Answer: Thank you for your suggestion, this has been included in the tables.

  1. Comment:  I recommend performing the same analyses for healthy individuals as well.

Answer: Thank you for your suggestion. This analysis has been run and the results have been included in a table in the supplementary materials.

  1. Comment: As far as I am aware, the cardiorespiratory optimal point, especially considering the reference authors used in the introduction, is the lowest VE/VO2 during a CPET. Surprisingly, authors used the lowest VE/VCO2

Answer: Thank you for identifying this. The cardiorespiratory optimal point was calculated using the lowest VE/VO2 during a CPET, this has now been more clearly described/ emphasised in the methods to avoid reader confusion.

  1. Comment: Please address medication use. Is this information available? Did the authors consider including it in the multiple regression? Such information is critical for HR recovery data. Please address this issue.

Answer: Thank you for your comment. Medication used has been included in a table in the supplementary material, however it was not included in the multiple linear regression as the authors feel that this is beyond the scope of this manuscript, but acknowledge that in future work this is something that should be explored.

  1. Comment: Please report critical information related to CPET “quality,” e.g., test termination criteria, peak RPE, peak RER, gas analyzer model/calibration/type (e.g., breath-by-breath vs mixed chamber), state pedaling frequency, room temperature (if controlled), etc.

Answer: Thank you for highlighting this missing information. Additional information about how the CPET was conducted and test termination criteria has been included in the methods section.

  1. Comment: Sample sizes should be included in all tables and figures.

Answer: Thank you for this suggestion, this has now been added to all tables.

  1. Comment: The test used to compare both groups - I suppose it was a T-test - is not mentioned in section 2.6. This information is important to interpret findings reported in Tables 1 and 2. Also, please clearly state how the authors adjusted for multiple testing.

Answer: Thank you for identifying this missing information. This information has now been added to the methods section.

  1. Comment: Often, the VE/VCO2 slope is presented solely as VE/VCO2, which, I believe, could be somewhat misleading. Please clarify this throughout the manuscript, clearly stating slope when appropriate, or, using another acronym like SlopeVE/VCO2.

Answer: Thank you for your comment. This has been addressed by amending this to SlopeVEVCO2 in the manuscript and tables.

  1. Comment: Revisit your limitations. Clearly address aspects such as the cross-sectional nature of the study, the inability to establish causation, the potential medication effect, the addictive nature of obesity, and T2D on the cardiovascular system.

Answer: Thank you for your comment. This has now been addressed in the limitations section of the discussion.

  1. Comment: Since Table 1 contains critical information and there is no limit to the number of tables and figures, I strongly recommend including Table 1 in the main manuscript rather than as supplementary material.

Answer: Thank you for your suggestion. This has now been amended and this table in now in the main tables document as opposed to in the supplementary material.

  1. Comment: Authors sometimes refer to the CPET variables analyzed as novel. I suggest different terms, as most of these variables have been investigated for more than ten years.

Answer: Thank you for your comment. The authors acknowledge that these variables are not novel however that term was used with regard to how they were being used in this context. However, we understand how this may be misleading and therefore different phrasing has been used throughout the manuscript.

  1. Comment: Suggestion: Include effect sizes in your report. Since you have a large sample size, p-values can sometimes be significant even when differences are small.

Answer: Thank you for your suggestion, effect sizes have now been included in the table.

  1. Comment: I recommend saying transmitral Doppler instead of transmittal.

Answer: Thank you for your suggestion. This has now been amended throughout.

  1. Comment: Please clarify if the βs used are standardized or not, and if they represent a change for every SD or unit.

Answer: The βs are not standardised.

  1. Comment: Adjust decimal places for a relevant clinical/functional interpretation. Resting HR could be 88 or 87.7 instead of 87.68. Same thing for age, BP, etc. (in tables)

Answer: Thank you for your suggestion. This has been addressed in all tables.

  1. Comment 17: Suggestion: Include a figure explaining your mechanistic rationale

Answer: Thank you for the suggestion. This has not been included as the aim of this manuscript was not to attempt to explore or identify mechanistic rationale behind the associations observed and thus, we feel that it is beyond the scope of this manuscript to include a figure attempting to explain the mechanistic rationale.

Reviewer 3 Report

Comments and Suggestions for Authors

REVIEW JFMK

I would like to thank the editorial team of the Journal of Functional Morphology and Kinesiology for the opportunity to review the manuscript entitled "Interrelationship Between Cardiopulmonary Exercise Testing Indices and Markers of Subclinical Cardiovascular Dysfunction in Those with Type 2 Diabetes". This is a highly relevant clinical and scientific topic with the potential to significantly contribute to the understanding of pathophysiological mechanisms and functional assessment strategies in individuals with type 2 diabetes. However, there is a clear need for substantial revisions to the current manuscript. I would like to reiterate that any personal remarks are intended solely to support the advancement of science and the professional development of the esteemed colleagues.

              First and foremost, there is a need to revise the structure of the manuscript, using the STROBE guidelines as a supporting tool to enhance data transparency. Therefore, each item listed in the STROBE checklist should be carefully reviewed to identify missing elements across the introduction, methods, results, and discussion sections.

Firstly, I suggest revising your title to include the type of study design being employed.

              In the Abstract, clearly state the research problem in the Background section, rather than providing only a generalized statement. Specify the dependent variables that will be used in the analyses when presenting the objectives. Up to the Results section, it remains unclear what the study actually intends to investigate. This becomes even more evident with the inclusion of the MRI procedure, which, up to that point, appears disconnected from the analysis. It is recommended to revise the Abstract, at least up to the Results section.

Your introduction appears well-structured and includes relevant information; however, it falls short in clearly identifying the research gap. Although you mention COP, thresholds, and heart rate decline, you fail to incorporate your dependent variables related to cardiovascular structure (e.g., Transthoracic Echocardiography). Therefore, I recommend adding an additional paragraph that contextualizes the other investigated variables, explicitly stating the research problem. Furthermore, at the end of the objectives paragraph, you should present your hypotheses, what you expected to find when designing this project (see STROBE). Please also include this citation in your introduction (https://doi.org/10.3390/app15073495).

Regarding your methods section, in the Participants subsection, please specify how participants were recruited, the recruitment period, exclusion criteria, and sample size calculation. These are non-negotiable elements. Add a section clearly presenting the primary and secondary dependent variables derived from the tests conducted. In your statistical analysis section, the predefined p-value threshold is not specified, nor are the statistical tests used to compare the different cardiopulmonary indices between patients with T2D and controls.

Your results are well presented; however, there appears to be a discrepancy between the objectives as initially stated and the outcomes reported in the results section. Consider the following: your primary comparison involves the indices obtained from patients with T2D versus controls, does this represent your primary outcome? Clearly define your intended primary outcome. If necessary, restructure your results section to include: general information (where you present baseline population characteristics), primary outcomes (e.g., regression analyses), and secondary outcomes (other relevant analyses). I believe your correlations could be more effectively visualized using a heatmap figure. Additionally, please present figures of the main regression analyses performed, including trend lines and the corresponding coefficients of determination.

Include a conclusion section that directly addresses your stated objectives. However, before doing so, clearly indicate in your objectives which are the primary and secondary outcomes.

THANKS!!!

Author Response

Overall comment

Thank you to the reviewer for the time taken to read this work and the incredibly helpful comments and suggested amendments. All comments and suggestions have been addressed with appropriate amendments made. The amended sections in the manuscript, tables, and figures have been highlighted in yellow for ease of identification.  We now feel that with the help of these comments and suggested amendments the quality of this manuscript has now been substantially improved. We would like to thank the reviewer again for the time taken to read and review this manuscript.

Please see below for responses to specific comments.

  1. Comment: there is a need to revise the structure of the manuscript, using the STROBE guidelines as a supporting tool to enhance data transparency. Therefore, each item listed in the STROBE checklist should be carefully reviewed to identify missing elements across the introduction, methods, results, and discussion sections.

Answer: Thank you for this suggestion. The authors used the STROBE checklist to identify missing elements throughout the manuscript.

  1. Comment: suggest revising your title to include the type of study design being employed.

Answer: Thank you for your suggestion. The title has been amended to include the study design.

  1. Comment: clearly state the research problem in the Background section, rather than providing only a generalized statement.

Answer: Thank you for your suggestion. The research gap this work is attempting to address has now been clearly acknowledged in the background.

  1. Comment: Specify the dependent variables that will be used in the analyses when presenting the objectives

Answer: Thank you for your comment. The dependant variables have been explicitly stated.

  1. Comment: Up to the Results section, it remains unclear what the study actually intends to investigate. This becomes even more evident with the inclusion of the MRI procedure, which, up to that point, appears disconnected from the analysis. It is recommended to revise the Abstract, at least up to the Results section.

Answer: Thank you for your comment. The abstract has now been revised to more clearly explain to the reader what the intention of the study is.

  1. Comment: Your introduction appears well-structured and includes relevant information; however, it falls short in clearly identifying the research gap

Answer: Thank you for your suggestion. The research gap this work is attempting to address has now been clearly acknowledged in the background.

  1. Comment: Although you mention COP, thresholds, and heart rate decline, you fail to incorporate your dependent variables related to cardiovascular structure (e.g., Transthoracic Echocardiography). Therefore, I recommend adding an additional paragraph that contextualizes the other investigated variables, explicitly stating the research problem.

Answer: Thank you for your comment. This has now been included in the discussion.

  1. Comment: at the end of the objectives paragraph, you should present your hypotheses, what you expected to find when designing this project (see STROBE). Please also include this citation in your introduction (https://doi.org/10.3390/app15073495).

Answer: thank you for your suggestion. A hypothesis has now been included in the introduction, and the suggested citation has been included.

  1. Comment: In the Participants subsection, please specify how participants were recruited, the recruitment period, exclusion criteria, and sample size calculation. These are non-negotiable elements.

Answer: Thank you for your comment. This was a secondary analysis of a already published trial. Therefore the authors is directed to the original manuscript for more detailed information regarding participant characteristic information including recruitment methods, exclusion criteria, and sample size calculation etc.

  1. Comment: Add a section clearly presenting the primary and secondary dependent variables derived from the tests conducted.

Answer: thank you for your comment. The results section has been amended to more clearly emphasise the dependant variables included in the analyses.

  1. Comment: In your statistical analysis section, the predefined p-value threshold is not specified, nor are the statistical tests used to compare the different cardiopulmonary indices between patients with T2D and controls.

Answer: Thank you for highlighting this. This information has now been added to the methods section.

  1. Comment: There appears to be a discrepancy between the objectives as initially stated and the outcomes reported in the results section

Answer: Thank you for your comment, The objectives have now been more clearly defined so that there is more flow between the objectives and outcomes.

  1. Comment: Consider the following: your primary comparison involves the indices obtained from patients with T2D versus controls, does this represent your primary outcome? Clearly define your intended primary outcome. If necessary, restructure your results section to include: general information (where you present baseline population characteristics), primary outcomes (e.g., regression analyses), and secondary outcomes (other relevant analyses). I believe your correlations could be more effectively visualized using a heatmap figure. Additionally, please present figures of the main regression analyses performed, including trend lines and the corresponding coefficients of determination.

Answer: thank you for your comment. Additional descriptions have eben included in the methods with regard to the outcomes. Regarding the additional figures the authors feel that this is beyond the scope of the manuscript and therefore have not been included. However, the tables have been amended to present more information, and more clearly, and effect sizes have also been added.

  1. Comment: Include a conclusion section that directly addresses your stated objectives. However, before doing so, clearly indicate in your objectives which are the primary and secondary outcomes.

Answer: Thank you for your comment. The conclusion has now been amended to include more information and to more clearly indicate objectives.

Round 2

Reviewer 2 Report

Comments and Suggestions for Authors

Dear authors,

Congratulations on your excellent work! You did a fantastic job addressing our queries, and your thorough responses demonstrate your dedication. It's clear that a lot of effort went into this. Thank you. Here are my observations.

Not sure if this was a .pdf type of issue, but Figure 1 is not presented.
State what type of effect size is being presented alongside the regression (Table 4). This could be done in the methods section or as part of the table legend.
As expected, healthy controls present different associations compared to the TD2+Obesity groups. Please consider addressing this succinctly.
Beta and p-values are slightly different between Supplementary Table 1 and Table 4. Please double-check.
Ve/VCO2 needs to be changed to SlopeVe/VCO2 in the supplementary table.
Table 1 states that the data are means +/- SD twice.

Author Response

General Comments:

I would like to thank the reviewers for taking the time to re-review this manuscript. The comments made by the reviewers are incredibly helpful and highlly valued. All the authors believe that this manuscript has been greatly improved thanks to the very insightful comments of the reviewers. Thank you for acknowledging that we have made appropriate amendments to the manuscript. Responses to comments relating to minor amendments are detailed below. In the documents any new changes are highlighted in pink for ease of identification.

Specific Comments

Comment 1. Not sure if this was a .pdf type of issue, but Figure 1 is not presented.

Response. Thank you for identifying this. I have ensured that figure 1 has been uploaded in the Zip folder with the manuscript.

Comment 2. State what type of effect size is being presented alongside the regression (Table 4). This could be done in the methods section or as part of the table legend.

Response. Thank you for identifying this missing information. This information has now been added to the figure legend.

Comment 3. As expected, healthy controls present different associations compared to the TD2+Obesity groups. Please consider addressing this succinctly.

Response. Thank you for suggesting this. This has now been addressed in the results.

Comment 4. Beta and p-values are slightly different between Supplementary Table 1 and Table 4. Please double-check.

Response. Thank you for your comment. I apologise if this wasn’t initially clear. Table 4 is the regression for people with T2D and obesity without adjustment for BMI (the original model), supplementary table 1 is with adjustment for BMI for both people with T2D and obesity and healthy volunteers. I have also tried to make this more clear in the text to avoid confusion.

Comment 5. Ve/VCO2 needs to be changed to SlopeVe/VCO2 in the supplementary table.

Response. Thank you for highlighting this typographical error. I have now amended this in the supplementary table.

Comment 6. Table 1 states that the data are means +/- SD twice.

Response. Thank you for highlighting this typographical error. I have now amended this in the supplementary table.

Reviewer 3 Report

Comments and Suggestions for Authors

Dear authors, thank you for considering the comments! I believe the manuscript has improved significantly.

Author Response

I would like to thank the reviewers for taking the time to re-review this manuscript. The comments made by the reviewers are incredibly helpful and highly valued. All the authors believe that this manuscript has been greatly improved thanks to the very insightful comments of the reviewers. Thank you for acknowledging that we have made appropriate amendments to the manuscript.